# Energy Management for Hybrid Electric Vehicles Using Safe Hybrid-Action Reinforcement Learning

Jinming Xu and Yuan Lin *

Shien-Ming Wu School of Intelligent Engineering, South China University of Technology,
Guangzhou 510641, China; wi_jinming@mail.scut.edu.cn
* Correspondence: yuanlin@scut.edu.cn

**Abstract:** Reinforcement learning has shown success in solving complex control problems, yet safety remains paramount in engineering applications like energy management systems (EMS), particularly in hybrid electric vehicles (HEVs). An effective EMS is crucial for coordinating power flow while ensuring safety, such as maintaining the battery state of charge within safe limits, which presents a challenging task. Traditional reinforcement learning struggles with safety constraints, and the penalty method often leads to suboptimal performance. This study introduces Lagrangian-based parameterized soft actor–critic (PASACLag), a novel safe hybrid-action reinforcement learning algorithm for HEV energy management. PASACLag utilizes a unique composite action representation to handle continuous actions (e.g., engine torque) and discrete actions (e.g., gear shift and clutch engagement) concurrently. It integrates a Lagrangian method to separately address control objectives and constraints, simplifying the reward function and enhancing safety. We evaluate PASACLag's performance using the World Harmonized Vehicle Cycle (901 s), with a generalization analysis of four different cycles. The results indicate that PASACLag achieves a less than 10% increase in fuel consumption compared to dynamic programming. Moreover, PASACLag surpasses PASAC, an unsafe counterpart using penalty methods, in fuel economy and constraint satisfaction metrics during generalization. These findings highlight PASACLag's effectiveness in acquiring complex EMS for control within a hybrid action space while prioritizing safety.

**Keywords:** hybrid electric vehicles; energy management strategy; safe reinforcement learning; hybrid action space; Lagrangian methods

**MSC:** 68T40





## 1. Introduction

Deep reinforcement learning (DRL) has achieved significant success in solving complex control problems, such as StarCraft [1], balloon navigation [2], Gran Turismo [3], and drone racing [4]. However, these impressive applications mainly focus on making specific decisions or outputting control commands to achieve optimal performance, without considering the safety of the system. In some safety-critical applications, such as energy management systems [5], autonomous driving [6], and robot control [7], ensuring safety is of paramount importance. Failure to guarantee safety in these contexts can lead to system damage or even casualties.

In recent years, there has been a significant surge in the development of a reinforcement learning-based energy management strategy (EMS) for hybrid electric vehicles (HEVs) [8]. In a typical HEV, energy can be sourced from multiple power sources, including internal combustion engine (ICE), electric motor, and energy storage systems such as batteries or supercapacitors. The EMS plays a vital role in coordinating the power flow between these sources based on various factors, including driving conditions, battery state of charge (SOC), power demand, and user preferences.

Traditionally, EMSs for HEVs rely on rule-based or optimization-based methods [9,10]. However, with the advent of reinforcement learning (RL) techniques, researchers have started exploring the application of RL algorithms in EMS development. RL is a branch of machine learning in which an agent learns to make optimal decisions through interactions with an environment. By applying RL, the EMS can adapt and optimize itself over time based on real-time data with a low computational cost. Utilizing RL for an EMS has the potential to improve fuel economy, reduce emissions, and enhance the overall performance of HEVs [8].

There have been various research efforts on applying RL to EMSs for HEVs. Hu et al. [11] proposed a deep Q-network (DQN)-based EMS for a parallel HEVs, where the control action is discretized output torque from the ICE. A simulation on ADVISOR showed that the equivalent fuel consumption of the proposed DQN-based EMS was 2.57% lower than that of the rule-based control strategy. To handle the continuous torque output directly, Liessner et al. [12] employed the deep deterministic policy gradient (DDPG) algorithm to the EMS. The researchers trained the model using stochastic driving cycles that shared similar characteristics with the testing conditions. The results showed that the DDPG strategy exhibited a fuel economy gap 0.9–1.7% higher than the dynamic programming (DP) approach. Additionally, Liu et al. [13] utilized the twin delayed deep deterministic policy gradient (TD3) algorithm to overcome the overestimation problem of the DDPG, and the results showed that the TD3-based EMS outperformed the DDPG-based EMS by 7.28% in terms of fuel consumption.

In addition to the consideration of a single action space in an EMS, researchers have also explored the application of reinforcement learning algorithms to handle more complex control tasks that involve both discrete and continuous actions. Li et al. [14] utilized a similar framework to the parameterized DDPG (PA-DDPG) proposed by Hausknecht et al. [15], where the weights of the discrete action driving mode and the continuous action engine torque are directly output by the actor network. Tang et al. [16] combined the DQN and DDPG, obtaining the gear ratio and engine throttle separately using the DQN and DDPG. Wang et al. [17] proposed the same algorithm as the parameterized DQN (P-DQN) [18] to control the clutch state and engine torque, in which the actor network first outputs the continuous engine torque for all clutch states, and then the Q-network selects the optimal clutch state with the highest Q-value.

Although the use of RL has demonstrated significant potential in the development of EMSs, it is crucial to acknowledge the challenges associated with ensuring the safety and dynamic constraints of RL-driven EMS applications. The exploratory nature and stochastic characteristics of RL algorithms can lead to the production of unreasonable actions, posing potential risks [8]. In the specific context of HEV systems, the initial stages of training and the assumption of unrestricted exploration may result in behaviors that cause damage to vehicle components and lead to hazardous outcomes [5]. For instance, an unbounded exploration in the action space can lead to excessive engine torque, causing overheating and damage to the engine. Similarly, excessive battery discharge resulting from uncontrolled exploration can lead to battery degradation and a reduced lifespan. Consequently, it is imperative to develop safe RL algorithms that effectively manage energy flow in HEVs while adhering to safety constraints.

A commonly employed strategy for handling safety constraints in RL is to introduce penalty terms into the reward function [19–21]. By incorporating these penalties, the agent receives negative rewards for violating the constraints during the learning process. Nevertheless, this formulation transforms the EMS problem into a multi-objective optimization problem, where the optimization objective encompasses both fuel economy and adherence to safety constraints. The balance between multiple objectives is regulated by the penalty coefficient, which poses a challenge as its determination is often non-trivial.

Fan et al. [22] added an additional safety layer to correct the continuous actions generated by the RL agent, which is trained using the penalty method in the first stage. The safety layer comprises a neural network that estimates how action changes affect the

safety sensitivity of the constrained variables by performing a first-order approximation of the constraints. Similarly, Zhang et al. [5] proposed a coach–learner framework, where the action output by the RL agent is first processed by a coach function, which determines whether the action satisfies all the constraints. If the action violates any of the constraints, a rule-based controller is activated to generate safe action. However, these methods require additional measures to ensure the safety of the system, which increases the complexity of the model and may reduce computational efficiency.

Lagrangian methods offer an alternative approach to address constraints in RL. This methodology involves initially modeling the system as a constrained Markov decision process (CMDP) [23], and subsequently converting the CMDP optimization problem into an unconstrained optimization problem by introducing Lagrange multipliers [24]. In particular, Chow et al. [24] proposed a gradient-based method for risk-constrained RL, which involves taking policy gradient steps on an objective that balances the trade-off between return and risk. A subsequent improvement to this approach was presented by Liang et al. [25], who enhanced the efficiency of the algorithm by incorporating off-policy trained dual variables. While these methodologies have demonstrated convergence towards constraint-satisfying policies, it is noteworthy that there has been limited research on the application of Lagrangian methods to hybrid action spaces thus far.

Recognizing these challenges, it is imperative to integrate principles of safe reinforcement learning to mitigate risks and ensure the reliability of EMS operations. Our work contributes to this endeavor by presenting a novel, safe hybrid-action reinforcement learning algorithm for HEV energy management. The proposed algorithm, named Lagrangian-based parameterized soft actor–critic (PASACLag), distinguishes itself by leveraging a distinctive composite action representation and Lagrangian methods to effectively manage both continuous and discrete variables concurrently. The principal contributions of this study can be summarized as follows:

1. The introduced energy management strategy, which utilizes a composite action representation method, showcases the ability to output two discrete actions, gear shift and clutch engagement, along with one continuous action, engine torque, simultaneously. This systematic optimization of both the continuous and discrete control variables is effective in improving the fuel economy of the HEV [26].

2. Extending beyond the hybrid action space, a novel safe hybrid-action reinforcement learning algorithm, PASACLag, is proposed for energy management in HEVs. PASACLag uniquely incorporates a Lagrangian method into this framework, facilitating the separate treatment of control objectives and constraints. This methodology not only simplifies the reward function but also bolsters the overall safety of the system.

The remainder of this paper is organized as follows: Section 2 provides the background on the constrained Markov decision process and hybrid action space. Section 3 introduces the proposed Lagrangian-based parameterized soft actor–critic. Section 4 presents the case study on an HEV. The experimental results are discussed in Section 5. Finally, Section 6 concludes the paper and discusses future work.

## 2. Preliminaries

We begin by introducing the basic concepts of the Markov decision process and extend it to the constrained Markov decision process. Following this, we provide a concise overview of the action space in reinforcement learning and introduce the concept of a hybrid action space.

### 2.1. Constrained Markov Decision Process

A Markov decision process (MDP) is a mathematical framework used to model sequential decision-making problems under uncertainty, which is commonly used in reinforcement learning [27]. It is represented by a tuple $(\mathcal{S}, \mathcal{A}, \mathcal{P}, \mathcal{R}, \gamma)$, where:

- $\mathcal{S}$ is the set of states representing the possible system configurations or states of the environment.

- $\mathcal{A}$ is the set of actions representing the available choices or actions that can be taken in each state.
- $\mathcal{P} : \mathcal{S} \times \mathcal{A} \times \mathcal{S} \to [0, 1]$ is the transition probability function. It defines the probability of transitioning from one state to another when taking a specific action.
- $\mathcal{R} : \mathcal{S} \times \mathcal{A} \to \mathbb{R}$ is the reward function. It assigns a numeric reward to each state–action pair, representing the immediate benefit or cost associated with taking a particular action in a specific state.
- $\gamma \in [0, 1]$ is the discount factor that weighs the importance of immediate rewards compared to future rewards. It determines the extent to which future rewards are considered in the decision-making process.

In a typical RL formulation, as depicted in Figure 1, the agent observes the state $s_t \in \mathcal{S}$ at each time step $t$, selects an action $a_t \in \mathcal{A}$ according to the policy $\pi$, and receives a reward $r_t \in \mathcal{R}$, and the next state $s_{t+1} \in \mathcal{S}$ according to the transition probability $\mathcal{P}$. The goal of the agent is to maximize the expected cumulative reward $R_t = \mathbb{E}_\pi \left[ \sum_{t=0}^{\infty} \gamma^t r_t(s_t, a_t) \right]$, and the optimal policy $\pi^*$ is defined as follows [27]:

$$\pi^* = \arg\max_\pi \mathbb{E}_\pi \left[ \sum_{t=0}^{\infty} \gamma^t r_t(s_t, a_t) \right]. \tag{1}$$

Therefore, define the action-value function $Q^\pi(s, a)$ as follows:

$$Q^\pi(s, a) = \mathbb{E}_\pi \left[ \sum_{t=0}^{\infty} \gamma^t r_t(s_t, a_t) \mid s_0 = s, a_0 = a \right], \tag{2}$$

where the optimal action at each state can be obtained by maximizing the action-value function $a^* = \arg\max_a Q^\pi(s, a)$.

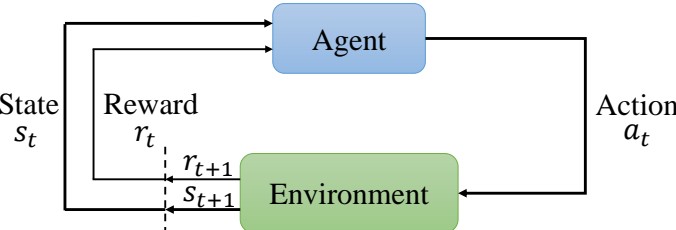

**Figure 1.** The Markov decision process for reinforcement learning.

The constrained Markov decision process extends the MDP framework by introducing constraints. It is represented by a tuple $(\mathcal{S}, \mathcal{A}, \mathcal{P}, \mathcal{R}, \gamma, c, \epsilon)$, where $c$ is the constraint function and $\epsilon$ is the constraint threshold. The objective of the agent is to maximize the expected cumulative reward $R_t$ while satisfying the constraint $c(s_t, a_t) \leq \epsilon$.

**Definition 1.** *A policy $\pi$ is considered optimal in a constrained Markov decision process if it maximizes the expected cumulative reward $R_t$ subject to a set of constraints $\mathcal{C}$. The optimal policy $\pi^*$ can be formulated as follows:*

$$\pi^* = \arg\max_\pi \mathbb{E}_\pi \left[ \sum_{t=0}^{\infty} \gamma^t r_t(s_t, a_t) \right]$$
$$\text{s.t.} \quad \mathbb{E}_\pi \left[ \sum_{t=0}^{\infty} \gamma^t c_t(s_t, a_t) \right] \leq \epsilon. \tag{3}$$

### 2.2. Hybrid Action Space

RL algorithms can be broadly classified into three categories based on the nature of the action space: discrete action space, continuous action space, and hybrid action space. A discrete action space consists of a finite set of distinct actions, represented as follows:

$$\mathcal{A} = \{a_1, a_2, \cdots, a_n\}, \tag{4}$$

where $a_i \in \mathcal{A}$ represents the $i$-th discrete action and $n$ denotes the total number of actions. The action space can also be extended to a multidimensional discrete action space:

$$\mathcal{A} = \mathcal{A}_1 \times \mathcal{A}_2 \times \cdots \times \mathcal{A}_n, \tag{5}$$

where $\mathcal{A}_i$ represents the $i$-th discrete action space. On the other hand, a continuous action space is a set of continuous actions represented as follows:

$$\mathcal{A} = \mathbb{R}^n, \tag{6}$$

where $n$ is the dimension of the action space.

The hybrid action space is a combination of discrete and continuous action spaces. A significant body of RL research treats this space as a parameterized action space [28]:

$$\mathcal{A} = \bigcup_{a \in \mathcal{A}_d} \{(a, \boldsymbol{x}) \mid \boldsymbol{x} \in \mathcal{X}_a\}, \tag{7}$$

where $\mathcal{A}_d = \{a_1, a_2, \ldots, a_k\}$ represents a finite set of discrete actions, and each discrete action $a_i \in \mathcal{A}_d$ is associated with a set of continuous parameters $\mathcal{X}_a \subseteq \mathbb{R}^{n_a}$, where $n_a$ represents the dimensionality. The action output typically follows a hierarchical relationship, with discrete actions at the top level and continuous parameters at the bottom level.

In this study, we investigate the concept of a parallel architecture to examine the hybrid action space. Within this framework, discrete and continuous actions are treated as separate entities without direct interdependence.

**Definition 2.** *A hybrid action space is a set of actions that comprises both discrete and continuous components. It is defined as follows:*

$$\mathcal{A} = \left\{ a = (\boldsymbol{a}^d, \boldsymbol{a}^c) \mid \boldsymbol{a}^d \in \mathcal{A}^d, \boldsymbol{a}^c \in \mathcal{A}^c \right\}, \tag{8}$$

*where $\mathcal{A}^d \subseteq \mathbb{N}^m$ represents a finite set of discrete action spaces, $\mathcal{A}^c \subseteq \mathbb{R}^n$ denotes a continuous action space, and $\boldsymbol{a}^d$ and $\boldsymbol{a}^c$ denote the discrete and continuous actions, respectively.*

## 3. Lagrangian-Based Parameterized Soft Actor–Critic

### 3.1. Entropy-Regularized Reinforcement Learning

Entropy is a quantity that measures the uncertainty of a random variable. Specifically, let $p(x)$ be the probability distribution of a random variable $x$, then the entropy of $x$ is defined as follows:

$$H(x) = -\mathbb{E}_{x \sim p(x)}[\log p(x)]. \tag{9}$$

In the context of reinforcement learning, we can use entropy to represent the uncertainty of the policy $\pi$ under a given state $s_t$ [29]:

$$H(\pi(\cdot|s_t)) = -\mathbb{E}_{a \sim \pi(\cdot|s_t)}[\log \pi(a|s_t)]. \tag{10}$$

The objective of entropy-regularized reinforcement learning is to maximize the expected cumulative reward $R_t$ while maximizing the entropy of the policy $\pi$, which will

encourage the agent to explore more actions and avoid premature convergence to suboptimal policies. This can be achieved by adding an entropy term to the objective function (1):

$$\pi^* = \arg\max_{\pi} \mathbb{E}_{\pi}\left[\sum_{t=0}^{\infty} \gamma^t(r_t(s_t, a_t) + \alpha H(\pi(\cdot|s_t)))\right], \tag{11}$$

where $\alpha > 0$ is the entropy regularization coefficient that controls the trade-off between the expected cumulative reward and the entropy of the policy.

With the introduction of the entropy term, the Bellman optimality equation of the action-value function $Q^{\pi}(s, a)$ can be rewritten as follows:

$$\begin{aligned} Q^{\pi}(s, a) &= \mathbb{E}_{s'\sim\mathcal{P}, a'\sim\pi}\left[r(s, a) + \gamma\big(Q^{\pi}(s', a') + \alpha H(\pi(\cdot|s')))\big)\right] \\ &= \mathbb{E}_{s'\sim\mathcal{P}, a'\sim\pi}\left[r(s, a) + \gamma\big(Q^{\pi}(s', a') - \alpha \log \pi(a'|s'))\big)\right]. \end{aligned} \tag{12}$$

*3.2. Parameterized Soft Actor–Critic*

The soft actor–critic (SAC) [29] is an off-policy actor–critic algorithm that combines the maximum entropy reinforcement learning framework with the neural network function approximator. While the original SAC algorithm is designed for continuous action spaces, we propose an extension of the SAC algorithm to handle hybrid action spaces, which we refer to as parameterized soft actor–critic (PASAC).

An actor–critic architecture consists of two neural networks: the actor network $\pi_{\theta}$ parameterized by $\theta$ and the critic network $Q_{\omega}$ parameterized by $\omega$. The actor network $\pi_{\theta}$ outputs the action $a_t$ given the state $s_t$, and the critic network $Q_{\omega}$ estimates the action-value function $Q(s, a)$. Through iterative updates to the actor and critic networks, the actor network learns to output the optimal action $a^*$ that maximizes the action-value function $Q(s, a)$, and the critic network learns to estimate the optimal action-value function $Q^*(s, a)$.

3.2.1. Learning the Critic Network

Update to the critic network is performed by minimizing the temporal difference error:

$$\mathcal{L}_Q(\omega) = \mathbb{E}_{s_t, a_t, r_t, s_{t+1}\sim\mathcal{D}}\left[(Q_{\omega}(s_t, a_t) - y_t)^2\right], \tag{13}$$

where $\mathcal{D}$ is the replay buffer that stores the transitions $(s_t, a_t, r_t, s_{t+1})$, and $y_t$ is the target given by the following:

$$y_t = r_t + \gamma(Q_{\omega^-}(s_{t+1}, a_{t+1}) - \alpha \log \pi_{\theta}(a_{t+1}|s_{t+1})). \tag{14}$$

To mitigate the overestimation problem that is known to degrade the performance of Q-learning algorithms [30], two target networks $Q_{\omega_1^-}$ and $Q_{\omega_2^-}$ are introduced to compute the target value $y_t$, each of which corresponds to a Q-network ($Q_{\omega_1}$ and $Q_{\omega_2}$, respectively). The minimum of the two target values is used as the final target, i.e.,

$$y_t = r_t + \gamma \min_{j=1,2}\left(Q_{\omega_j^-}(s_{t+1}, a_{t+1}) - \alpha \log \pi_{\theta}(a_{t+1}|s_{t+1})\right). \tag{15}$$

3.2.2. Learning the Actor Network

The actor network is updated by maximizing the expected cumulative reward $R_t$ while maximizing the entropy of the policy $\pi$. The objective function is defined as follows:

$$\begin{aligned} \mathcal{J}_{\pi}(\theta) &= \mathbb{E}_{s_t\sim\mathcal{D}, a_t\sim\pi_{\theta}}[Q_{\omega}(s_t, a_t) - \alpha \log \pi_{\theta}(a_t|s_t)] \\ &= \mathbb{E}_{s_t\sim\mathcal{D}, a_t\sim\pi_{\theta}}\left[\min_{j=1,2} Q_{\omega_j}(s_t, a_t) - \alpha \log \pi_{\theta}(a_t|s_t)\right]. \end{aligned} \tag{16}$$

Here, the minimum of the two Q-networks is utilized as the estimation of the action-value $Q(s_t, a_t)$. For a continuous action space, the actor network outputs the mean $\mu_\theta(s_t)$ and the standard deviation $\sigma_\theta(s_t)$ of a Gaussian distribution, and the action $a_t$ is sampled from the Gaussian distribution $\mathcal{N}(\mu_\theta(s_t), \sigma_\theta(s_t))$. However, the sampling process is not differentiable, so the reparameterization trick is used to obtain a differentiable function. Specifically, the action $a_t$ is sampled from a standard Gaussian distribution $\mathcal{N}(0, 1)$, and then the action $a_t$ is transformed to $\mu_\theta(s_t) + \sigma_\theta(s_t) \odot \delta$, where $\delta \sim \mathcal{N}(0, 1)$ is a random noise vector and $\odot$ denotes the element-wise product. Therefore, the objective function of the actor network can be rewritten as follows:

$$\mathcal{J}_\pi(\theta) = \mathbb{E}_{s_t \sim \mathcal{D}, \delta \sim \mathcal{N}(0,1)} \left[ \min_{j=1,2} Q_{\omega_j}(s_t, \tilde{a}_t) - \alpha \log \pi_\theta(\tilde{a}_t | s_t) \right], \quad \tilde{a}_t = \mu_\theta(s_t) + \sigma_\theta(s_t) \odot \delta. \tag{17}$$

3.2.3. Practical Implementation for Hybrid Action Space

As introduced in Section 2.2, the hybrid action space is defined as follows:

$$\mathcal{A} = \left\{ a = (\boldsymbol{a}^d, \boldsymbol{a}^c) \mid \boldsymbol{a}^d \in \mathcal{A}^d, \boldsymbol{a}^c \in \mathcal{A}^c \right\}, \quad \mathcal{A}^d \subseteq \mathbb{N}^m, \quad \mathcal{A}^c \subseteq \mathbb{R}^n. \tag{18}$$

To represent this hybrid action space in a linear fashion suitable for neural network outputs, we flatten the multi-dimensional discrete and continuous components into a single composite vector:

$$a = (\boldsymbol{a}^d, \boldsymbol{a}^c) \rightarrow [a_1^d, a_2^d, \dots, a_i^d, \dots, a_m^d, a_1^c, a_2^c, \dots, a_i^c, \dots, a_n^c]. \tag{19}$$

Here, each discrete action $a_i^d$ is an element selected from a finite set of possible actions $\mathcal{A}_i^d$. The total number of possible combinations for the discrete actions is, thus, the product of the number of feasible choices for each action, represented as $\prod_{i=1}^{m} d_i$.

We configure the output dimension of the actor network to accommodate both sets of actions, setting it at $\mathfrak{m} + n$, where $\mathfrak{m} = \prod_{i=1}^{m} d_i$. The output from the actor network, denoted as $\mathfrak{a}$, is as follows:

$$\mathfrak{a} = [\mathfrak{a}_1, \mathfrak{a}_2, \dots, \mathfrak{a}_\mathfrak{m}, \mathfrak{a}_{\mathfrak{m}+1}, \dots, \mathfrak{a}_{\mathfrak{m}+n}]. \tag{20}$$

For discrete actions $\boldsymbol{a}^d$, we apply the `argmax` function to the segments of $\mathfrak{a}$ corresponding to each discrete action's dimension, effectively selecting the action with the highest utility from the feasible action set. The continuous actions $\boldsymbol{a}^c$ are directly based on the remaining elements of $\mathfrak{a}$, which represent real-valued actions.

*3.3. Lagrangian-Based Parameterized Soft Actor–Critic*

In this study, we employ the Lagrangian method as our chosen approach to address the constrained optimization problem in the context of CMDP within the PASAC framework. Lagrangian methods are a well-established and widely used technique for handling constrained optimization problems. By introducing a Lagrangian multiplier, we can transform the original constrained optimization problem into an unconstrained optimization problem [31]. This transformation simplifies the RL reward function by decoupling the learning of reward and cost networks. The dual-network approach, supported by the Lagrange multiplier, enables the RL agent to learn policies that simultaneously maximize rewards while respecting constraints, which may represent safety requirements.

The constrained optimization problem of the CMDP in PASAC can be formulated as follows:

$$\pi^* = \arg\max_{\pi} \mathbb{E}_{\pi}\left[\sum_{t=0}^{\infty} \gamma^t (r_t(s_t, a_t) + \alpha H(\pi(\cdot|s_t)))\right]$$

$$\text{s.t.} \quad \mathbb{E}_{\pi}\left[\sum_{t=0}^{\infty} \gamma^t c_t(s_t, a_t)\right] \leq \epsilon.$$

(21)

The optimal policy $\pi^*$ can then be obtained by solving the corresponding unconstrained optimization problem:

$$(\pi^*, \lambda^*) = \arg\min_{\lambda \geq 0} \max_{\pi} \mathcal{L}(\pi, \lambda)$$

$$= \arg\min_{\lambda \geq 0} \max_{\pi} \left( \mathbb{E}_{\pi}\left[\sum_{t=0}^{\infty} \gamma^t (r_t(s_t, a_t) + \alpha H(\pi(\cdot|s_t)))\right] \right.$$

$$\left. - \lambda\left( \mathbb{E}_{\pi}\left[\sum_{t=0}^{\infty} \gamma^t c_t(s_t, a_t)\right] - \epsilon\right)\right),$$

(22)

where $\lambda$ denotes the Lagrangian multiplier, dynamically adjusted to penalize the policy proportionally to the cost incurred, thus enforcing the constraints.

In practice, the optimal policy $\pi^*$ can be obtained by alternating between the following two steps:

1.  Solve the unconstrained optimization problem (22) to determine a feasible policy $\pi$.
2.  Increase the Lagrangian multiplier $\lambda$ until the constraint is satisfied.

In the initial step, the unconstrained optimization problem (22) can be addressed through the PASAC algorithm by incorporating the Lagrangian multiplier $\lambda$ into the objective function (17):

$$\mathcal{J}_{\pi}(\theta) = \mathbb{E}_{s_t \sim \mathcal{D}, a_t \sim \pi_{\theta}}\left[\min_{j=1,2} Q_{\omega_j}(s_t, \tilde{a}_t) - \alpha \log \pi_{\theta}(\tilde{a}_t|s_t) - \lambda(Q_{\phi}(s_t, \tilde{a}_t) - \epsilon)\right], \quad (23)$$

where $Q_{\phi}$ is the cost network that estimates the constraint violations. It follows a similar update rule to the critic network using the temporal difference error. The Lagrangian multiplier $\lambda$ is treated as a constant, and the actor network is updated through gradient descent:

$$\theta \leftarrow \theta - \beta \nabla_{\theta} \mathcal{L}_{\pi}(\theta), \quad (24)$$

where $\beta$ is the step size. It is noteworthy that when $\lambda$ assumes large values, the update in Equation (24) may induce excessively large changes to the parameter $\theta$, potentially leading to instability. Motivated by the approach presented in Ref. [32], we introduce a rescaled objective function to ensure consistent step sizes:

$$\theta \leftarrow \theta - \beta \frac{1}{1+\lambda} \nabla_{\theta} \mathcal{L}_{\pi}(\theta). \quad (25)$$

In the subsequent step, the Lagrangian multiplier $\lambda$ is updated through gradient descent:

$$\lambda \leftarrow \lambda - \eta(\epsilon - Q_{\phi}(s_t, a_t)), \quad (26)$$

where $\eta$ represents the step size.

The pseudocode of the PASACLag algorithm is presented in Algorithm 1.

---

**Algorithm 1:** Pseudocode of the PASACLag Algorithm

---

**Input:** Gradient stepsizes $\{\alpha, \beta, \xi, \eta\} \geq 0$, empty replay buffer $\mathcal{D}$, minibatch size $B$, discount factor $\gamma$, soft target update rate $\tau$, cost limit $\epsilon$, initial Lagrangian multiplier $\lambda$.

1  Initialize the Q-network $Q_{\omega_1}, Q_{\omega_2}$, actor network $\mu_\theta$, and cost network $Q_\phi$ with random weights $\omega_1, \omega_2, \theta, \phi$;

2  Initialize the target networks $\omega_1^- \leftarrow \omega_1, \omega_2^- \leftarrow \omega_2, \phi^- \leftarrow \phi$;

3  **for** $t = 1 : T$ **do**

4  $\quad$ Observe state $s_t$ and select action $a_t$ according to policy $\pi_\theta$;

5  $\quad$ Execute action $a_t$ and observe reward $r_t$ and the next state $s_{t+1}$;

6  $\quad$ Store transition $(s_t, a_t, r_t, c_i, s_{t+1})$ in replay buffer $\mathcal{D}$;

7  $\quad$ If $s_{t+1}$ is a terminal state, reset the environment;

8  $\quad$ Sample $B$ transitions $\{(s_i, a_i, r_i, c_i, s_{i+1})\}_{i \in [B]}$ from $\mathcal{D}$;

9  $\quad$ Compute the target value for the Q networks

$$y_i = r_i + \gamma \min_{j=1,2} \left( Q_{\omega_j^-}(s_{i+1}, a_{i+1}) - \alpha \log \pi_\theta(a_{i+1}|s_{i+1}) \right);$$

10  $\quad$ Update the two Q-networks by one step of gradient descent

$$\omega_j \leftarrow \omega_j - \alpha \nabla_{\omega_j} \frac{1}{|B|} \sum_{i \in [B]} \left( Q_{\omega_j}(s_i, a_i) - y_i \right)^2, \qquad \text{for } j = 1, 2;$$

11  $\quad$ Update the cost network by one step of gradient descent

$$\phi \leftarrow \phi - \xi \nabla_\phi \frac{1}{|B|} \sum_{i \in [B]} \left( Q_\phi(s_i, a_i) - \left( c_i + \gamma Q_{\phi^-}(s_{i+1}, a_{i+1}) \right) \right)^2;$$

12  $\quad$ Update the actor network by one step of gradient ascent

$$\theta \leftarrow \theta + \frac{\beta}{1 + \lambda} \nabla_\theta \frac{1}{|B|} \sum_{i \in [B]} \left( \min_{j=1,2} Q_{\omega_j}(s_i, \tilde{a}_i) - \alpha \log \pi_\theta(\tilde{a}_i|s_i) - \lambda \left( Q_\phi(s_i, \tilde{a}_i) - \epsilon \right) \right),$$

$\quad$ where $\tilde{a}_i = \mu_\theta(s_i) + \sigma_\theta(s_i) \odot \delta, \quad \delta \sim \mathcal{N}(0, 1)$;

13  $\quad$ Update the target networks with

$$\omega_j^- \leftarrow \tau \omega_j + (1 - \tau) \omega_j^-, \qquad \text{for } j = 1, 2$$
$$\phi^- \leftarrow \tau \phi + (1 - \tau) \phi^-;$$

14  $\quad$ Update the Lagrangian multiplier $\lambda$ by one step of gradient descent

$$\lambda \leftarrow \lambda - \eta \frac{1}{|B|} \sum_{i \in [B]} \left( \epsilon - Q_\phi(s_i, a_i) \right);$$

15  **end**

---

## 4. Case Study

### 4.1. Modeling of the HEV Powertrain

The host vehicle studied in this work was a parallel hybrid electric vehicle [33] with an internal combustion engine, an electric motor, an automatic clutch, and a 6-speed automated manual transmission, as depicted in Figure 2. The main parameters of the vehicle are provided in Table 1.

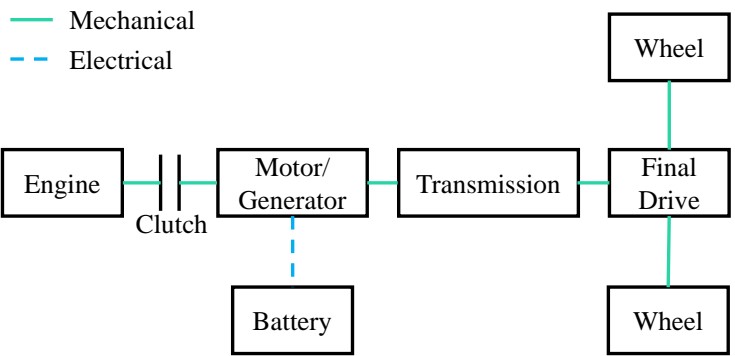

**Figure 2.** The parallel hybrid powertrain architecture.

**Table 1.** Main parameters of the HEV.

| Symbol | Value | Unit | Description |
| --- | --- | --- | --- |
| $\omega_{max}$ | 250 | rad/s | Maximum motor/engine angular velocity |
| $T_{b,max}$ | 6000 | Nm | Maximum mechanical brake torque |
| $i_d$ | 4.11 | - | Final drive gear ratio |
| $\eta_f$ | 0.931 | - | Final drive efficiency |
| $\eta_g$ | 0.931 | - | Transmission efficiency |
| $Q_{max}$ | 6.5 | Ah | Battery capacity |
| $N_b$ | 112 | - | Number of battery cells |
| $m$ | 5000 | kg | Vehicle mass |
| $A$ | 6.73 | $m^2$ | Vehicle cross-sectional area |
| $r$ | 0.5715 | m | Tyre radius |
| $\mu$ | 0.01 | - | Rolling resistance coefficient |
| $\rho$ | 1.1985 | $kg/m^3$ | Air density |
| $C_d$ | 0.65 | - | Air drag coefficient |
| $\theta$ | 0 | rad | Road grade |

The vehicle dynamics are described by the following equation:

$$T_w = \left( \frac{\rho C_d A v^2}{2} + \mu m g \cos \theta + m g \sin \theta + m a \right) r, \tag{27}$$

where $T_w$ is the torque acting on the wheel, $\rho$ is the air density, $C_d$ is the air drag coefficient, $A$ is the vehicle's cross-sectional area, $v$ is the vehicle velocity, $\mu$ is the coefficient of rolling resistance, $m$ is the vehicle mass, $g$ is the gravitational acceleration, $\theta$ is the road grade, $a$ is the vehicle acceleration, and $r$ is the tire radius. The shaft speed of the wheel is related to the vehicle velocity as follows:

$$\omega = \frac{v}{r} i_d i_g, \tag{28}$$

where $i_d$ is the final drive gear ratio and $i_g$ is the transmission gear ratio. Since the motor is directly connected to the transmission, the motor speed is equal to the transmission speed.

The torque on the wheel is transmitted from the engine and motor through the clutch and transmission, which can be described as follows:

$$\frac{T_w + T_b}{i_d i_g \eta_d \eta_g} = \varrho T_{e,d} + T_m, \tag{29}$$

where $T_b$ is the mechanical brake torque, $i_d$ is the final drive gear ratio, $i_g$ is the transmission gear ratio, $\eta_d$ is the final drive efficiency, $\eta_g$ is the transmission efficiency, $T_{e,d}$ is the engine driving torque, $T_m$ is the motor torque, and $\varrho$ is the clutch engagement status, which is a binary variable that takes the value of 1 when the clutch is engaged and 0 when the clutch is disengaged.

The engine torque is separated into two parts: the driving torque $T_{e,d}$, which propels the vehicle, and the idle torque $T_{e,i}$, which powers the auxiliary systems. The engine speed is governed by the following equation:

$$\omega_e = \begin{cases} \omega, & \text{if } \varrho = 1, \\ \omega_{e,i}, & \text{if } \varrho = 0, \end{cases} \tag{30}$$

where $\omega$ is the motor angular velocity, and $\omega_{e,i}$ is the engine idle angular velocity. The engine fuel map and the maximum engine torque are shown in Figure 3a, where the engine fuel consumption rate $\dot{m}_f = f_f(\omega, T_e)$ is a function of the engine angular velocity $\omega$ and the engine torque $T_e$. The motor efficiency map and the maximum/minimum motor torque are shown in Figure 3b, where the motor efficiency $\eta_m = f_\eta(\omega, T_m)$ is a function of the motor angular velocity $\omega$ and the motor torque $T_m$.

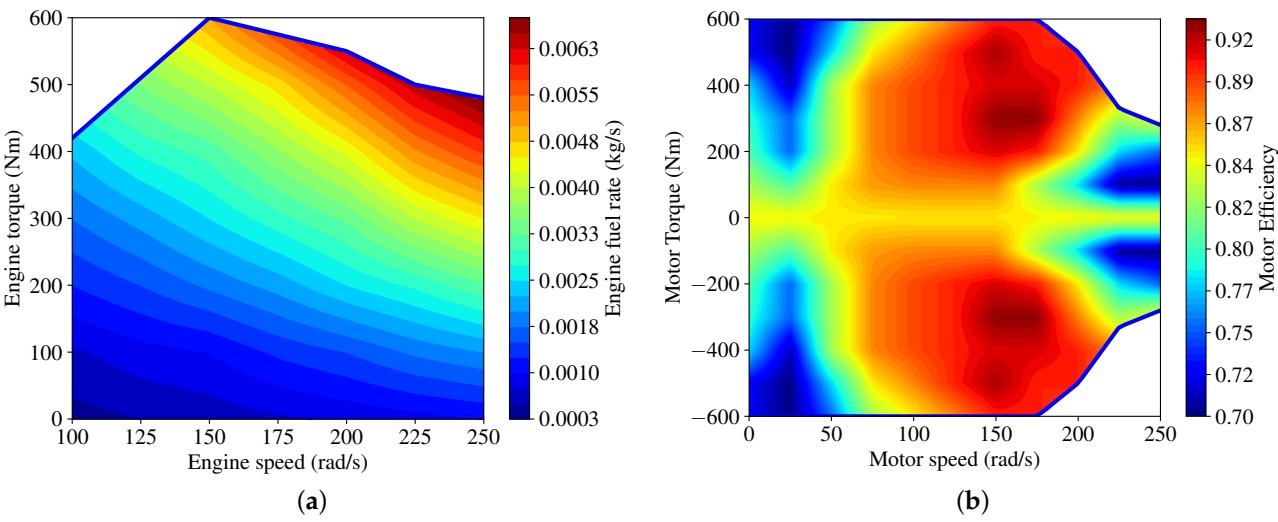

(**a**)           (**b**)

**Figure 3.** (**a**) The engine fuel map and the maximum engine torque (blue line). (**b**) The motor efficiency map and the maximum/minimum motor torque (blue line).

The battery state of charge is calculated using the following:

$$\dot{SOC} = -\frac{I_b}{Q_{max}} = -\frac{E - \sqrt{E^2 - 4R_b T_m \omega \eta_m^{\text{sgn}(-T_m)}}}{2R_b Q_{max}}, \tag{31}$$

where $Q_{max}$ is the battery capacity; the open-circuit voltage $E$ of the battery cell and the battery internal resistance $R_b$ are denoted as functions of the SOC, as shown in Figure 4.

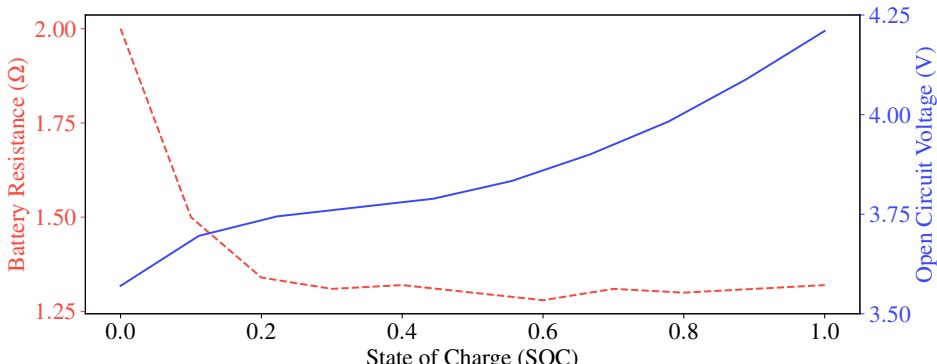

**Figure 4.** The open-circuit voltage and resistance of the battery cell.

### 4.2. Problem Formulation

The goal of an HEV energy management system is to determine the optimal power split between the engine and the motor to minimize the fuel consumption while satisfying the vehicle dynamics and the battery SOC constraints. Therefore, the optimization problem can be formulated as follows:

$$\min \sum_{t=0}^{T_{cyc}} \dot{m}_f(t) \tag{32}$$

$$\text{s.t.} \quad \begin{cases} 0 \le \omega \le \omega_{max}, \\ 0 \le T_e \le T_{e,max}, \\ T_{m,min} \le T_m \le T_{m,max}, \\ 0 \le T_b \le T_{b,max}, \\ SOC_{min} \le SOC \le SOC_{max}, \end{cases} \tag{33}$$

where $T_0$ and $T_{cyc}$ denote the initial and final times of the driving cycle.

### 4.3. Implementation of PASACLag

Based on the problem formulation, the key components of the PASACLag algorithm are described as follows.

**State:** The observation at each time step is defined as follows:

$$s = [v, a, T_w, SOC, i_g, \varrho]. \tag{34}$$

Here, the vehicle velocity $v$, the vehicle acceleration $a$, and the required torque $T_w$ provide the power demand information of the vehicle, and the SOC, the transmission gear ratio $i_g$, and the clutch engagement status $\varrho$ provide the information of the powertrain.

**Action:** The hybrid action is defined as follows:

$$a = [T_e, \varsigma, \varrho], \tag{35}$$

where $\varsigma$ represents a shift command with three options: upshift, downshift, and no shift.

To comprehensively ascertain the power distribution between the engine and the motor, the system takes into account two distinct scenarios. Firstly, when the demanded torque $T_w$ assumes a negative value, the system prioritizes the utilization of regenerative braking to recharge the battery, with any surplus torque being supplied by mechanical braking. In cases where the regenerative braking capacity falls short of meeting the power requirements, the remaining torque is supplemented through mechanical braking. Secondly, in situations where the demanded torque $T_w$ is positive, the braking torque is set to zero, and the required torque is jointly supplied by the engine and the motor.

**Reward:** Since the objective of the energy management system is to minimize the fuel consumption, the reward function is defined as the negative of the fuel consumption rate:

$$r = -\dot{m}_f. \tag{36}$$

**CMDP formulation:** Recapping the constraints specified in Equation (33), the engine torque constraint can be directly achieved through the definition of the action space, and the braking torque constraint can be implemented during the execution of the action. However, direct enforcement of the remaining constraints is not feasible due to system dynamics. To address these constraints within the CMDP framework uniformly, an indicator function is introduced to evaluate the violation of each constraint individually:

$$c = \mathbb{I}(\omega) + \mathbb{I}(SOC) + \mathbb{I}(T_m), \tag{37}$$

$$\mathbb{I}(*) = \begin{cases} 0, & \text{if } * \in [*_{min}, *_{max}], \\ 1, & \text{otherwise.} \end{cases} \tag{38}$$

Thus, the CMDP formulation of the problem is as follows:

$$\pi^* = \arg\max_{\pi} \mathbb{E}_{\pi} \left[ \sum_{t=0}^{\infty} \gamma^t (r(s_t, a_t) + \alpha H(\pi(\cdot|s_t))) \right]$$

$$\text{s.t.} \quad \mathbb{E}_{\pi} \left[ \sum_{t=0}^{\infty} \gamma^t c(s_t, a_t) \right] \le 0. \tag{39}$$

It is worth noting that the constraint threshold is set to zero, indicating an ideal scenario where no constraint violation occurs throughout the entire driving cycle.

## 5. Experiments

### 5.1. Experimental Setup

The proposed algorithm was evaluated on the urban driving segment of the World Harmonized Vehicle Cycle (WHVC), as depicted in Figure 5. This segment contains frequent speed changes, stops, and idling, which makes it challenging to achieve optimal and safe energy management.

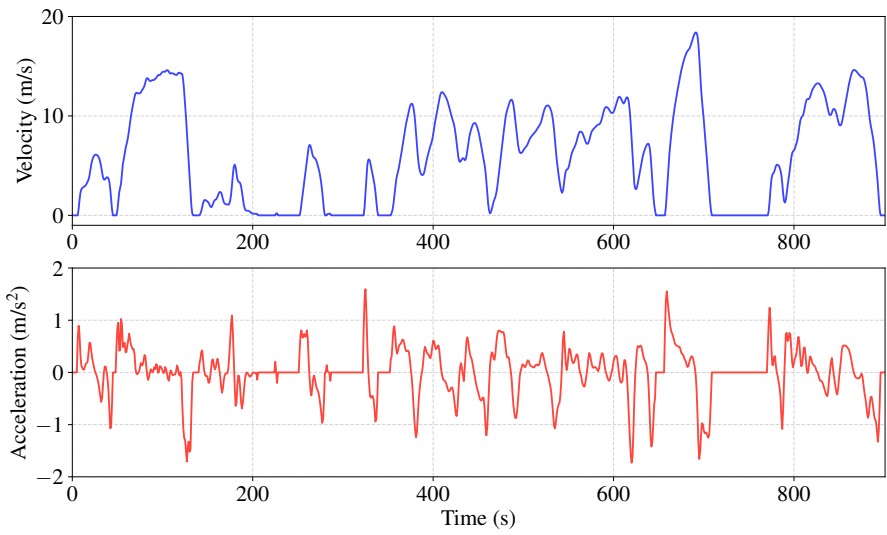

**Figure 5.** The urban driving segment of World Harmonized Vehicle Cycle.

The performance of the proposed algorithm was evaluated through a comparative analysis with DP and PASAC. DP employs the MATLAB toolbox developed by Sundström et al. [34], which enables the imposition of constraints on variables and the direct definition of the loss function as the fuel consumption of the system. With fine-grained discretization, DP can be regarded as the optimal solution to the problem. The state variable selected was the SOC, with 400 grids ranging from 0.4 to 0.8. The input variables chosen were the same as those used in the RL algorithm, i.e., the engine torque $T_e$, the shift command $\varsigma$, and the clutch engagement status $\varrho$. Notably, the continuous engine torque was discretized with an interval of 5 Nm.

As outlined in Section 3.2, PASAC is introduced. Since PASAC cannot directly handle safety constraints, a penalty term is incorporated into the reward function to penalize constraint violations. The updated reward function is expressed as follows:

$$r = -\dot{m}_f - p_\omega - p_{T_m} - p_{SOC},$$ (40)

$$p_\omega = 10\tilde{p}, \quad \text{if } \omega > \omega_{max},$$ (41)

$$p_{T_m} = 10\tilde{p}, \quad \text{if } T_m > T_{m,max},$$ (42)

$$p_{SOC} = \begin{cases} 0, & \text{if } SOC_{min} \leq SOC \leq SOC_{max}, \\ 10\tilde{p} \cdot \frac{|SOC - SOC_{max}|}{1 - SOC_{max}}, & \text{if } SOC > SOC_{max}, \\ 10\tilde{p} \cdot \frac{|SOC - SOC_{min}|}{SOC_{min}}, & \text{if } SOC < SOC_{min}, \end{cases}$$ (43)

where $\tilde{p}$ represents a reference penalty value set to be the intermediate value of the fuel consumption rate.

The environment was implemented using the Safety-Gymnasium framework [35], which is an extension of the OpenAI Gym framework [36] designed for safety-critical applications in RL. Upon each reset of the environment, the initial SOC was randomly drawn from a uniform distribution within the range of $SOC_{min}$ to $SOC_{max}$. Additionally, the start time of the driving cycle was sampled from a uniform distribution between $T_0$ and $T_{cyc}$ to introduce variability in the initial conditions of each episode. The RL training process involved a total of 3000 episodes, with each episode corresponding to the length of a single driving cycle. The actor networks and the critic networks of the RL algorithms were instantiated as fully connected neural networks, each comprising two hidden layers. Specifically, PASACLag featured 64 neurons in each layer, while PASAC boasted 128 neurons in each layer. The optimization of network parameters was executed using the Adam optimizer [37], which is recognized for its efficiency in dealing with large-scale and non-convex optimization problems. We set a learning rate of 0.0003 for both the actor and critic networks, a value that was determined empirically to yield stable convergence during preliminary testing. All the experiments were conducted on a workstation equipped with an AMD Ryzen 9 5950X CPU and an NVIDIA RTX 3080Ti GPU.

*5.2. Learning Ability Assessment*

The training results of the RL algorithms are depicted in Figure 6. For robust result reliability, each algorithm underwent three training iterations using different random seeds. Each seed generated a unique random initial state for the environment. Initially, both PASAC and PASACLag displayed a rapid increase in average reward during the training process, accompanied by relatively high costs. This phenomenon indicated a tendency to violate constraints in pursuit of a higher reward at the beginning. However, as training progressed, the average reward for both algorithms gradually converged to a stable value, with simultaneous reduction in costs to a lower level. This suggested that both algorithms successfully learned a policy that adhered to the specified constraints. Comparing the two, PASACLag exhibited a slightly higher average reward than PASAC, and its cost could converge to zero. This implies that PASACLag not only learned a policy satisfying the constraints but also demonstrated superior performance compared to PASAC.

The results in Table 2 provide insights into the computational efficiency of different strategies. DP does not have a training time as it focuses on solving a known problem optimally without a dedicated training phase. The average execution time represents the computational effort per step. PASACLag and PASAC exhibit significantly lower execution times compared to DP, indicating faster decision-making and policy updates. However, it is worth noting that PASACLag has a slightly slower execution time (0.94 milliseconds per step) compared to PASAC (0.91 milliseconds per step). This difference can be attributed to PASACLag having an additional cost network to learn, which adds some computational overhead.

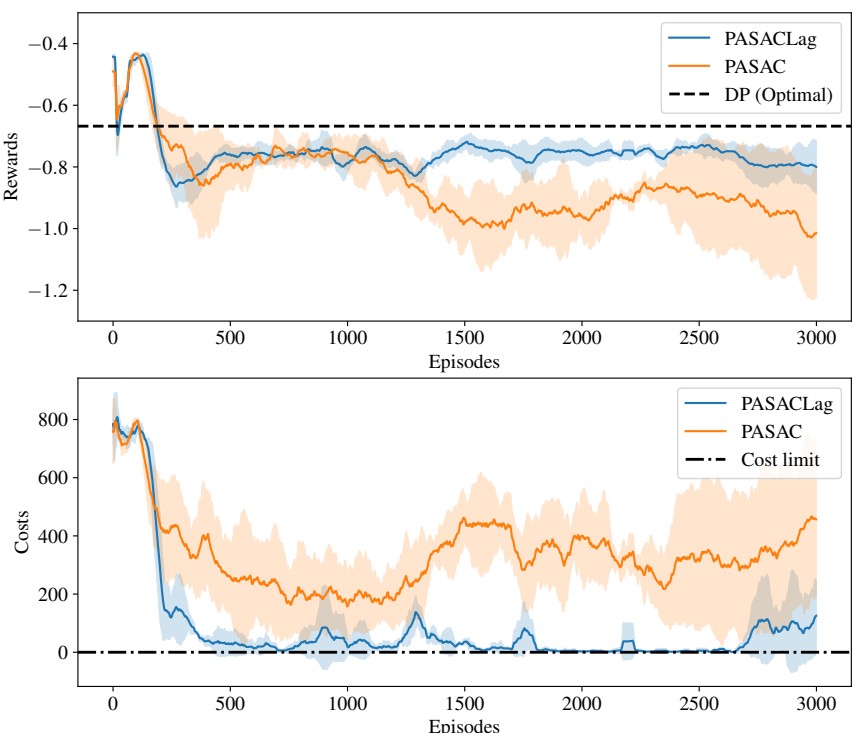

**Figure 6.** Training results of the RL algorithms on the WHVC driving cycle. The solid lines represent the average reward of the three different seeds, and the shaded areas represent the standard deviation. Note that the penalty term is excluded from the reward plot for PASAC.

**Table 2.** The training time and average execution time comparison.

| Strategy | Training Time (h) | Execution Time (ms) |
|----------|-------------------|---------------------|
| DP | N/A | 592,077 |
| PASACLag | 4.90 | 0.94 |
| PASAC | 3.89 | 0.91 |

*5.3. Energy Management Performance*

The SOC trajectories of the three algorithms are shown in Figure 7. Both PASAC and PASACLag show a similar SOC change trend to the optimal DP solution and successfully maintain the SOC within the desired range. The discrepancies in the SOC trajectories are mainly attributed to the different constraint satisfaction strategies and the learning dynamics of the RL algorithms.

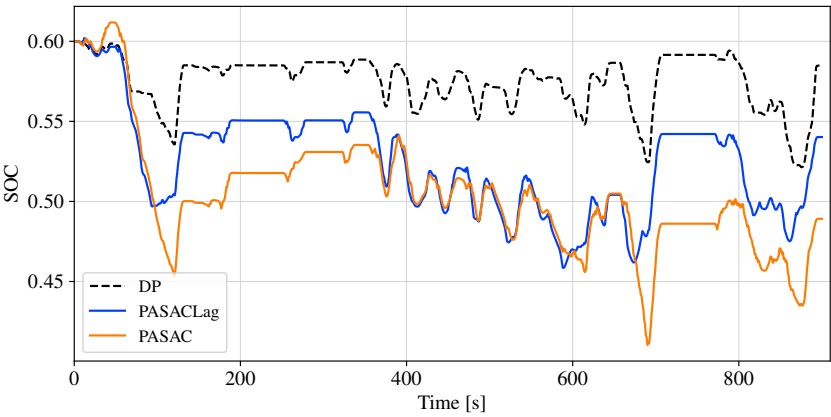

**Figure 7.** SOC trajectories for WHVC driving cycle.

To ensure a fair comparison of fuel consumption performance, the final SOC of the three methods needs to be corrected. The GB/T 19754 standard provides a systematic conversion method [38]. The aim of the correction is to determine an equivalent factor that translates the difference in SOC into a corresponding fuel consumption adjustment. The trained model used different initial SOC values for simulation. A total of six groups were tested, with three groups sampled from $[SOC_0, SOC_{max}]$ for initial SOC, and the other three groups sampled from $[SOC_{min}, SOC_0]$. The differences in SOC ($\Delta SOC$) and fuel consumption were recorded and used to obtain the equivalent factor, which is represented by the least square fitting line shown in Figure 8. The positive slope of the line represents the equivalent coefficient between $\Delta SOC$ and the fuel consumption. If the $\Delta SOC$ in a cycle is positive, it indicates that electrical energy is being consumed, and additional fuel consumption should be added to the originally simulated fuel consumption to ensure a fair comparison of fuel consumption.

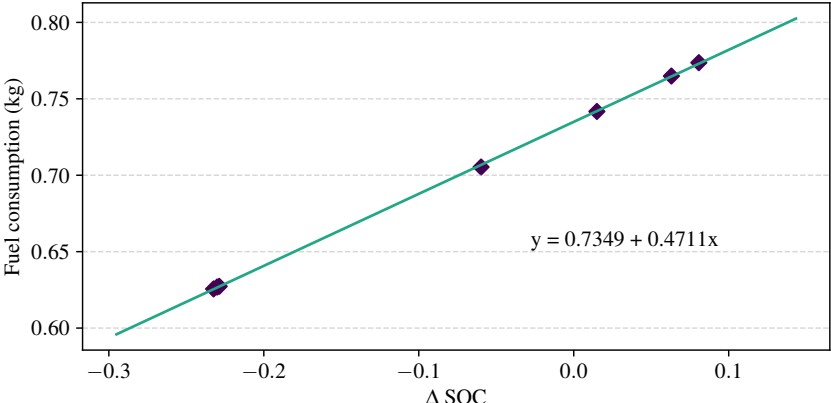

**Figure 8.** The least square fitting line of the data.

The final SOC using the DP strategy serves as the reference in the comparison. The results of the comparison are summarized in Table 3. Across the three different seeds, the average fuel consumption of PASACLag and PASAC is 9.32% and 11.11% higher than that of DP, respectively. Notably, PASACLag exhibits no constraint violations throughout the entire driving cycle, while PASAC averages one constraint violation.

**Table 3.** The energy management performance comparison when trained and tested with the same driving cycle WHVC ($SOC_0 = 0.6$).

| Strategy | Seed | $SOC_o$ | $Fuel_o$ (kg) | $SOC_e$ | $Fuel_e$ (kg) | Gap | $\psi$ |
|---|---|---|---|---|---|---|---|
| DP | - | 0.5858 | 0.6678 | 0.5858 | 0.6678 | 0 | 0 |
| PASACLag | 0 | 0.6394 | 0.7526 | 0.5858 | 0.7292 | 9.19% | 0 |
|  | 1 | 0.7221 | 0.8138 | 0.5858 | 0.7521 | 12.62% | 0 |
|  | 2 | 0.4892 | 0.6979 | 0.5858 | 0.7447 | 11.51% | 0 |
| PASAC | 0 | 0.4751 | 0.6871 | 0.5858 | 0.7329 | 9.74% | 1 |
|  | 1 | 0.5468 | 0.7076 | 0.5858 | 0.7290 | 9.17% | 0 |
|  | 2 | 0.5402 | 0.7054 | 0.5858 | 0.7282 | 9.05% | 0 |

$SOC_o$: the original unconverted final SOC; $Fuel_o$: the original unconverted fuel consumption; $SOC_e$: the converted final SOC; $Fuel_e$: the converted equivalent fuel consumption; $\psi$: the number of constraint violations.

Figure 9 illustrates the power distribution and hybrid action output of DP and RL for WHVC. The continuous engine torque and motor torque of RL fall within the desired range. Regarding discrete actions, the gear shift and clutch engagement of RL exhibit a sequence similar to that of DP, with slightly more frequent transitions. This observation

suggests that the RL algorithm successfully learned a judicious power distribution strategy within the hybrid action space.

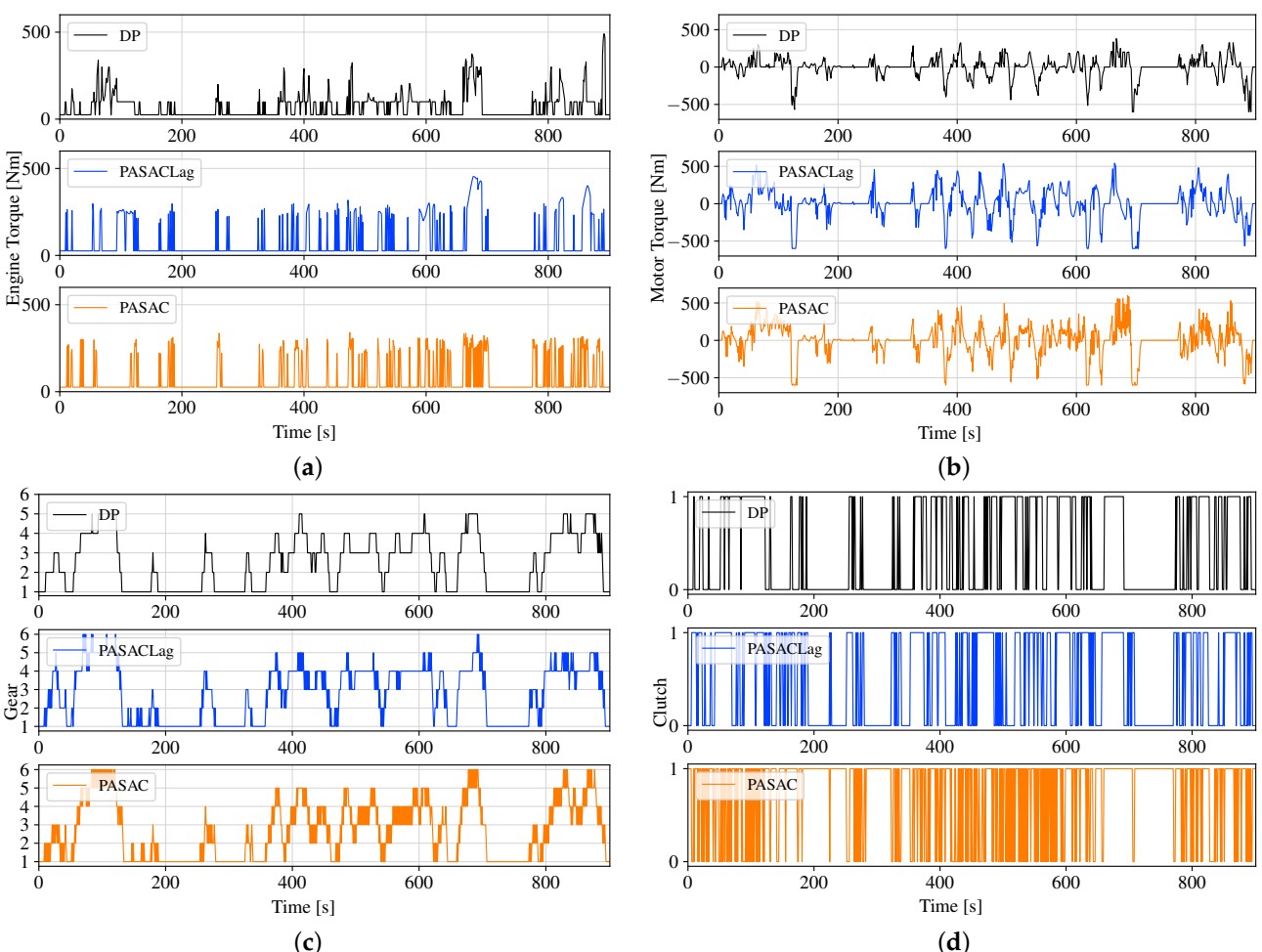

**Figure 9.** The power distribution and hybrid action output of DP and RL for WHVC. (**a**) Engine torque. (**b**) Motor torque. (**c**) Gear shift sequence. (**d**) Clutch state (0 = disengaged, 1 = engaged).

*5.4. Generalization Validation*

To assess the generalization capability and robustness of the proposed algorithm, the trained model underwent testing on four additional driving cycles: HD-UDDS, CHTC-LT, JE05, and HHDDT. The summarized results are presented in Table 4. Evidently, across all four driving cycles, the fuel consumption and constraint violations of PASACLag consistently outperformed those of PASAC, underscoring the superior performance of PASACLag in terms of both fuel economy and compliance with constraints. The results have significant implications for real-world applications, as they demonstrate the algorithm's potential for delivering consistent performance and efficiency across a wide range of driving scenarios. Nevertheless, the granular analysis of violations reveals that PASACLag incurs a modest number of infractions related to motor torque and SOC. This suggests that, while effective, the algorithm does not achieve flawless generalization across new driving cycles.

**Table 4.** Generalization results of the new driving cycles.

| Cycle | Strategy | Fuel$_e$ (kg) | Gap | Number of Constraint Violations | | |
|---|---|---|---|---|---|---|
| | | | | $\psi_{T_m}$ | $\psi_{\omega}$ | $\psi_{soc}$ |
| HD-UDDS | DP | 1.2408 | 0 | - | - | - |
| | PASACLag | 1.3058 | 5.24% | 17 | 0 | 0 |
| | PASAC | 1.3453 | 8.42% | 2 | 8 | 60 |
| CHTC-LT | DP | 1.9460 | 0 | - | - | - |
| | PASACLag | 2.0861 | 7.20% | 4 | 0 | 0 |
| | PASAC | 2.1112 | 8.49% | 1 | 4 | 192 |
| JE05 | DP | 1.7387 | 0 | - | - | - |
| | PASACLag | 1.8644 | 7.23% | 4 | 0 | 0 |
| | PASAC | 1.8914 | 8.78% | 2 | 3 | 156 |
| HHDDT | DP | 4.9813 | 0 | - | - | - |
| | PASACLag | 5.1889 | 4.17% | 0 | 0 | 44 |
| | PASAC | 5.2949 | 6.29% | 0 | 32 | 1352 |

$T_m$: motor torque; $\omega$: shaft angular velocity; *soc*: SOC.

## 6. Conclusions

This paper introduces a novel RL algorithm, PASACLag, for HEV energy management. PASACLag's innovative composite action representation adeptly manages the intricacies of continuous and discrete control actions, such as engine torque adjustments and gear shifts. By incorporating a Lagrangian approach to distinctly tackle control objectives and constraints, the algorithm not only simplifies the reward structure but also significantly improves the safety of the learning process.

Our comprehensive evaluation demonstrates that PASACLag consistently outperforms its predecessor, PASAC, by achieving better fuel economy and adhering better to operational constraints, with a less than 10% increase in fuel consumption compared to the benchmark set by dynamic programming across various driving cycles.

The significance of these findings lies in PASACLag's ability to balance complex control demands within a hybrid action space, ensuring both efficiency and safety, which are critical in the context of HEV energy management. Furthermore, the execution speed of the algorithm bolsters its practicality for real-time applications.

Looking ahead, we are committed to advancing this research by conducting hardware-in-the-loop simulations to validate the feasibility and durability of PASACLag in real-world scenarios. Additionally, we aim to refine the algorithm's generalization capabilities to further enhance its adaptability and performance across diverse and unpredictable driving conditions.

**Author Contributions:** Conceptualization, Y.L.; data curation, J.X.; formal analysis, J.X.; funding acquisition, Y.L.; investigation, J.X.; methodology, J.X. and Y.L.; project administration, Y.L.; resources, Y.L.; software, J.X.; supervision, Y.L.; validation, J.X. and Y.L.; visualization, J.X.; writing—original draft, J.X.; writing—review and editing, J.X. and Y.L. All authors have read and agreed to the published version of the manuscript.

**Funding:** This work was supported in part by Guangzhou Basic and Applied Basic Research Program under Grant 2023A04J1688, and in part by South China University of Technology faculty start-up fund.

**Data Availability Statement:** The data presented in this study are available on request.

**Conflicts of Interest:** The authors declare no conflicts of interest.

## Abbreviations

The following abbreviations are used in this manuscript:

| | |
|---|---|
| CMDP | Constrained Markov decision process |
| DDPG | Deep deterministic policy gradient |
| DP | Dynamic programming |
| DQN | Deep Q-network |
| DRL | Deep reinforcement learning |
| EMS | Energy management strategy |
| HEV | Hybrid electric vehicle |
| ICE | Internal combustion engine |
| MDP | Markov decision process |
| P-DQN | Parameterized deep Q-network |
| PASAC | Parameterized soft actor–critic |
| PASACLag | Lagrangian-based parameterized soft actor–critic |
| RL | Reinforcement learning |
| SAC | Soft actor–critic |
| SOC | State of charge |
| TD3 | Twin delayed deep deterministic policy gradient |
| WHVC | World Harmonized Vehicle Cycle |
| $\pi$ | Policy |
| $s$ | State |
| $a$ | Action |
| $r$ | Reward |
| $c$ | Constraint |
| $Q(s,a)$ | Action-value function |
| $\mathcal{A}$ | Action space |
| $H(\pi)$ | Entropy of the policy |
| $\mathcal{L}$ | Loss function |
| $\mathcal{J}$ | Objective function |

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
