# Peer review of "Energy Management for Hybrid Electric Vehicles Using Safe Hybrid-Action Reinforcement Learning"

_mathematics, doi:10.3390/math12050663_

Round 1
Reviewer 1 Report
Comments and Suggestions for Authors
The topic of the research is interesting one. Note the following review comments:
1. write the full term for WHVC in abstract
2. In the abstract, lines 12 to 13, and in the conclusion, "reveals fuel consumption less than 10% higher": gives bit confusion while reading; try to re-write the sentence correctly
3. In section 2.1, the procedure for Markov decision process need to be provided with appropriate references for the statements
4. Most of the equations are not properly acknowledged with appropriate references
5. It needs to give abbreviations for the variables used in the equations
6. In Section 3.1, assign numbers and consider as subheadings "learning the critic network," "learning the actor network," and "practical implementation for hybrid action space":
7. As the authors mentioned, there is a similar SOC change trend for both PASACLag and PASAC, but till close to 400 s, there is more difference in SOC trends with both methods. Similarly, after 700 s, more differences between the SOC levels of PASACLag and PASAC. State the reason for that.
8. The authors need to write further details about the the tools used for analysis in methodology
9. Write down the limitations of the proposed PASACLag strategy.
Comments on the Quality of English LanguageNeeds to check minor spell checks and grammatical errors throughout the manuscript
Reviewer 2 Report
Comments and Suggestions for Authors
Energy management systems play an important role in hybrid electric vehicles. To improve their effectiveness, this paper proposes a hybrid-action reinforcement learning algorithm named Lagrangian-based parameterized soft actor-critic (PASACLag). The following are my suggestions/comments on this work.
1. In line no. 5, it is recommended to first mention the issues with the state-of-the-art works and then introduce the proposed method.
2. Include more details on the evaluation setup, such as the specific metrics used, the duration of simulations, and any other relevant parameters in the abstract.
3. Explicitly state the motivation for focusing on safety in energy management systems for hybrid electric vehicles. This will help readers understand the significance of your work.
4. Explain how the composite action representation effectively handles both continuous and discrete actions in a more detailed manner.
5. Elaborate on why the Lagrangian method is chosen and how it contributes to simplifying the reward function and enhancing safety.
6. Please provide insights into the computational efficiency of PASACLag compared to alternative methods.
7. Insert a flow diagram of reinforcement learning to understand the concept clearly.
8. It is suggested to add nomenclature at the beginning of the introduction due to more usage of abbreviations and also advised to check all the terms are defined properly.
9. Provide insights into the generalization capabilities of PASACLag on different driving cycles. Highlight any challenges faced and how the algorithm overcomes them.
10. Authors should indicate the reason behind a "reinforcement learning" based approach because scaling to complex environments with high-dimensional state and action spaces poses computational and memory challenges.
11. The conclusion section currently looks too similar to the abstract. Rewrite it to focus on the main takeaways of the research (Summarize the main findings, discuss the significance of findings).
Round 2
Reviewer 2 Report
Comments and Suggestions for Authors
The paper has been revised well.